# Healthy lifestyle and cognitive decline in middle-aged and older adults residing in 14 European countries

Mikaela Bloomberg [1] ✉, Graciela Muniz-Terrera[2], Laura Brocklebank[1] & Andrew Steptoe [1]

Studies examining lifestyle and cognitive decline often use healthy lifestyle indices, making it difficult to understand implications for interventions. We examined associations of 16 lifestyles with cognitive decline. Data from 32,033 cognitively-healthy adults aged 50-104 years participating in prospective cohort studies of aging from 14 European countries were used to examine associations of lifestyle with memory and fluency decline over 10 years. The reference lifestyle comprised not smoking, no-to-moderate alcohol consumption, weekly moderate-plus-vigorous physical activity, and weekly social contact. We found that memory and fluency decline was generally similar for non-smoking lifestyles. By contrast, memory scores declined up to 0.17 standard deviations (95% confidence interval= 0.08 – 0.27) and fluency scores up to 0.16 standard deviations (0.07 – 0.25) more over 10 years for those reporting smoking lifestyles compared with the reference lifestyle. We thus show that differences in cognitive decline between lifestyles were primarily dependent on smoking status.

With the global population aging, aging-related neurodegenerative conditions such as dementia have become an increasingly pressing public health concern. As the most prevalent cause of dementia, Alzheimer's disease is characterized by progressive cognitive decline and a lengthy preclinical period during which neuropathology accumulates in the decades before diagnosis[1]. Given the paucity of effective treatments for Alzheimer's disease and the long preclinical period, it is important to identify modifiable factors that might slow cognitive decline in the decades before disease onset to delay clinical symptoms.

Many lifestyle and behavioral factors—including smoking, alcohol consumption, frequency and intensity of participation in physical activity, and social contact—have been identified as potential determinants of the pace of cognitive ageing [2] and risk of dementia[3]. While it is well-established that these behavioral factors are independently associated with cognitive health, they may also combine to influence cognitive ageing trajectories, particularly as healthy behaviors tend to cluster: individuals with healthy habits for one lifestyle domain tend to

have other healthy habits as well[4]. As such, there is a growing body of literature examining combinations of healthy behaviors and their associations with cognitive health. A recent systematic review[5] identified 16 longitudinal studies that examined combinations of behavioral factors and their associations with cognitive ageing trajectories. Many of these studies used healthy lifestyle indices[6–12] to suggest more healthy behaviors are correlated with better cognitive outcomes, but this method does not allow for examination of individual behaviors, and also generally assumes all healthy behaviors contribute equally to cognitive function. This reduces the relevancy of these studies as a basis for designing interventions. Other studies have insufficient sample sizes to examine multiple combinations of behaviors[6–8,10,13–16], include follow-up periods less than 10 years[6,7,9,10,13,17,18], focus on cardiovascular risk factors only[19], dichotomize cognitive function[20], or do not examine domain-specific trajectories[21].

Given these limitations, it is necessary to undertake further analyses to understand how behavioral factors combine to influence

[1]Department of Behavioural Science and Health, University College London, London, UK. [2]Heritage College of Osteopathic Medicine, Ohio University, Athens, Ohio, USA. ✉e-mail: mikaela.bloomberg.19@ucl.ac.uk

cognitive decline, particularly in sufficiently large study populations to examine combinations of behaviors with precision. Furthermore, because prodromal dementia symptoms can impact behavior years before clinical diagnosis[22], it is important to assess behavioral risk factors for cognitive decline in populations who are not yet experiencing disease-related cognitive symptoms.

To address these considerations and elucidate relationships between lifestyle and cognitive decline, we used up to 15 years of longitudinal data from over 32,000 adults aged 50–104 years without cognitive impairment or dementia in cohorts from 14 European countries. We examined associations of 16 lifestyles (comprising smoking, alcohol, physical activity, and social contact habits) with 10-year cognitive decline in two cognitive domains that show deterioration with dementia (episodic memory and verbal fluency). In this work, we show that associations between lifestyle and cognitive decline mostly depended on smoking habits; cognitive decline was faster for smoking lifestyles and generally similar for non-smoking lifestyles.

## Results
### Sample characteristics
Of 9131 respondents aged 50 years and older participating in wave 2 of the English Longitudinal Study of Ageing (ELSA), 1,088 (11.9%) were missing more than one behavior at baseline and therefore did not have missing behavior data imputed, 1,821 (19.9%) either reported dementia diagnosis or had cognitive scores suggesting cognitive impairment, 5 (<0.1%) were missing cognitive scores at all waves of follow-up, and 24 (0.3%) were missing covariates; these respondents were excluded leading to 6193 ELSA respondents being included in analyses (Figure S1).

Of 32,657 respondents aged 50 years and older participating in wave 2 of the Survey of Health, Ageing and Retirement in Europe (SHARE), 125 (0.4%) had alcohol top-coded at 70 drinks per day, 595 (1.8%) were missing more than one behavior, 5,962 (18.3%) reported dementia diagnosis or had cognitive scores suggesting cognitive impairment, 109 (0.3%) were missing cognitive scores at all waves, and 26 (<0.1%) were missing covariates; these respondents were excluded leading to 25,840 SHARE respondents being included in analyses (Figure S2). The pooled analytic sample included 32,033 ELSA and SHARE participants in total.

Participant characteristics at baseline are presented in Table 1 and are shown in men and women separately in Tables S1, S2. Participants who did not smoke were more likely to be older at baseline, female, more educated, and wealthier than those who smoked ($p < 0.001$ for all); they were also more likely to report diagnoses of diabetes, cancer, cardiovascular conditions, high blood pressure ($p < 0.001$ for all), and high cholesterol ($p = 0.002$), and less likely to report psychiatric conditions ($p = 0.017$). Participants who reported no-to-moderate alcohol consumption were more likely to be older at baseline, male, less educated, and less wealthy than those who reported heavy alcohol consumption ($p < 0.001$ for all) and were more likely to report diabetes, cardiovascular conditions, high cholesterol, high blood pressure ($p < 0.001$ for all), and psychiatric conditions ($p = 0.002$). Participants reporting at least weekly MVPA tended to be younger at baseline, male, more educated, and wealthier than those reporting less-than-weekly MVPA, and were less likely to report all chronic conditions ($p < 0.001$ for all). Participants reporting at least weekly social contact tended to be younger at baseline, female, were more educated and wealthier than those reporting less-than-weekly social contact ($p < 0.001$ for all), were less likely to have diabetes, cardiovascular conditions, high blood pressure ($p < 0.001$ for all), high cholesterol ($p = 0.012$), and more likely to have cancer ($p < 0.001$) and lung disease ($p = 0.047$).

Median follow-up durations were similar for those reporting more and less recommendation-compliant behavior for each of the four behaviors. The median follow-up period for both those who reported smoking and those who did not smoke was 8 years (interquartile range:

0–12). The median follow-up period for those reporting heavy alcohol consumption was 10 years (2–13) compared with 8 (0–12) for those reporting no-to-moderate alcohol consumption. The median follow-up period for those with weekly MVPA was 10 years (2–13) compared with 8 (0–12) for less-than-weekly MVPA. Finally, the median follow-up for those reporting weekly social contact was 10 years (4–13) compared with 8 (0–12) for those reporting less-than-weekly social contact.

The number of participants with each lifestyle is reported in Table 2, with the number of participants remaining in the analytic sample at each study wave reported in Table S3. Median follow-up durations were similar for all lifestyles (between 7.5 and 10 years; Table 2), with the exception of two lifestyles: 1) smoking, no-to-moderate alcohol consumption, less-than-weekly MVPA, and less-than-weekly social contact (median=6 years, IQR = 0–11) and 2) non-smoking, no-to-moderate alcohol consumption, less-than-weekly MVPA, and less-than-weekly social contact (median=6 years; IQR = 0–11). Mean baseline cognitive performance differed between lifestyles, ranging from −0.33 SD (SD = 1.04) for memory or −0.32 SD (0.99) for fluency in the non-smoking, no-to-moderate alcohol consumption, less-than-weekly MVPA, less-than-weekly social contact lifestyle to 0.33 SD (0.94) for memory in the non-smoking, heavy alcohol consumption, weekly MVPA, weekly social contact lifestyle or 0.32 SD (1.04) for fluency in the smoking, heavy alcohol consumption, weekly MVPA, weekly social contact lifestyle (Tables S4, S5).

### Independent association of behaviors with cognitive decline
After adjustment for covariates, non-smoking and no-to-moderate alcohol consumption were independently associated with a slower decline in both memory and fluency (Table S6). Non-smokers had memory scores that declined 0.08 standard deviations (SD; 95% confidence interval [CI]=0.05 to 0.12) less than smokers over 10 years ($p < 0.001$), whilst those reporting no-to-moderate alcohol consumption declined 0.04 SD (0.01 to 0.06) less than those reporting heavy alcohol consumption ($p = 0.005$). Non-smokers' fluency scores declined 0.08 SD (0.05 to 0.11) less than smokers ($p < 0.001$), and those reporting no-to-moderate alcohol declined 0.03 SD (0.01 to 0.06) less than those reporting heavy alcohol consumption ($p = 0.010$). There were no independent associations between memory decline and MVPA ($p = 0.575$) or social contact ($p = 0.464$) or between fluency decline and social contact ($p = 0.349$). There was weak evidence of an association between weekly MVPA and faster fluency decline (difference in decline over 10 years = −0.03 SD [−0.05 to 0.00]; $p = 0.040$).

### Lifestyle and cognitive decline
Memory decline over 10 years from age 65 for each lifestyle is shown in Table 3. After adjustment for covariates, participants reporting the most recommendation compliant lifestyle (referred to as the reference lifestyle) had memory scores that declined −0.20 SD (−0.23 to −0.16) during 10 years of follow-up.

Differences in memory decline over 10 years between the reference lifestyle and the other lifestyles are shown in Fig. 1. Participants who reported smoking and less-than-weekly social contact had memory scores that declined faster than the reference lifestyle, regardless of alcohol consumption or physical activity habits. Those reporting heavy alcohol consumption had memory scores that declined −0.14 SD ([−0.23 to −0.04] with less-than-weekly MVPA; $p = 0.005$) or −0.17 ([−0.27 to −0.08] with weekly MVPA; $p < 0.001$) more than the reference lifestyle over 10 years. Those reporting no-to-moderate alcohol consumption declined −0.10 SD (−0.18 to −0.02) more than the reference lifestyle with less-than-weekly MVPA ($p = 0.014$) or −0.14 (−0.22 to −0.05) more with weekly MVPA ($p = 0.002$).

In general, participants reporting smoking and weekly social contact also had faster memory decline than the reference lifestyle, but these differences did not always reach statistical significance. Participants reporting smoking, heavy alcohol consumption, less-than-

**TABLE 1 | Characteristics of the pooled analytic sample at study baseline (N = 32,033)**

| | Non-smoking | | | No-to-moderate alcohol consumption | | | Weekly MVPA | | | Weekly social contact | | |
|---|---|---|---|---|---|---|---|---|---|---|---|---|
| | No N = 5900 | Yes N = 26133 | P-value | No N = 11101 | Yes N = 20932 | P-value | No N = 17837 | Yes N = 14196 | P-value | No N = 16902 | Yes N = 15131 | P-value |
| Age at baseline, mean (SD) | 60.5 (8.1) | 65.3 (9.9) | <0.001 | 61.8 (8.5) | 65.8 (10.1) | <0.001 | 66.5 (10.3) | 61.8 (8.4) | <0.001 | 65.0 (10.2) | 63.8 (9.2) | <0.001 |
| Gender | | | | | | | | | | | | |
| Men | 3099 (52.5) | 11218 (42.9) | <0.001 | 4743 (42.7) | 9574 (45.7) | <0.001 | 7416 (41.6) | 6901 (48.6) | <0.001 | 7915 (46.8) | 6402 (42.3) | <0.001 |
| Women | 2801 (47.5) | 14915 (57.1) | | 6358 (57.3) | 11358 (54.3) | | 10421 (58.4) | 7295 (51.4) | | 8987 (53.2) | 8729 (57.7) | |
| Education | | | | | | | | | | | | |
| Below upper secondary | 2544 (43.1) | 11831 (45.3) | <0.001 | 3999 (36.0) | 10376 (49.6) | <0.001 | 8970 (50.3) | 5405 (38.1) | <0.001 | 8604 (50.9) | 5771 (38.1) | <0.001 |
| Upper secondary | 2326 (39.4) | 9323 (35.7) | | 4439 (40.0) | 7210 (34.4) | | 6124 (34.3) | 5525 (38.9) | | 5709 (33.8) | 5940 (39.3) | |
| Tertiary | 1030 (17.5) | 4979 (19.1) | | 2663 (24.0) | 3346 (16.0) | | 2743 (15.4) | 3266 (23.0) | | 2589 (15.3) | 3420 (22.6) | |
| Standardized wealth, mean (SD) | −0.07 (0.90) | 0.06 (1.10) | <0.001 | 0.07 (1.19) | 0.02 (0.99) | <0.001 | −0.01 (0.98) | 0.09 (1.16) | <0.001 | 0.00 (1.03) | 0.07 (1.10) | <0.001 |
| Diagnosis of: | | | | | | | | | | | | |
| Diabetes | 471 (8.0) | 2652 (10.1) | <0.001 | 817 (7.4) | 2306 (11.0) | <0.001 | 2145 (12.0) | 978 (6.9) | <0.001 | 1930 (11.4) | 1193 (7.9) | <0.001 |
| Cancer | 302 (5.1) | 1737 (6.6) | <0.001 | 686 (6.2) | 1353 (6.5) | 0.333 | 1333 (7.5) | 706 (5.0) | <0.001 | 999 (5.9) | 1040 (6.9) | <0.001 |
| Lung disease | 690 (11.7) | 2887 (11.0) | 0.160 | 1187 (10.7) | 2390 (11.4) | 0.052 | 2467 (13.8) | 1110 (7.8) | <0.001 | 1831 (10.8) | 1746 (11.5) | 0.047 |
| Cardiovascular condition | 735 (12.5) | 4621 (17.7) | <0.001 | 1374 (12.4) | 3982 (19.0) | <0.001 | 3894 (21.8) | 1462 (10.3) | <0.001 | 3005 (17.8) | 2351 (15.5) | <0.001 |
| Psychiatric condition | 528 (8.9) | 2090 (8.0) | 0.017 | 833 (7.5) | 1785 (8.5) | 0.002 | 1690 (9.5) | 928 (6.5) | <0.001 | 1396 (8.3) | 1222 (8.1) | 0.564 |
| High cholesterol | 1273 (21.6) | 6135 (23.5) | 0.002 | 2420 (21.8) | 4988 (23.8) | <0.001 | 4446 (24.9) | 2962 (20.9) | <0.001 | 4004 (23.7) | 3404 (22.5) | 0.012 |
| High blood pressure | 1802 (30.5) | 10502 (40.2) | <0.001 | 3795 (34.2) | 8509 (40.7) | <0.001 | 7854 (44.0) | 4450 (31.3) | <0.001 | 6754 (40.0) | 5550 (36.7) | <0.001 |

Data shown are N (%) unless otherwise indicated. Pearson's x² test used for categorical variables; t-test used for continuous variables. Two-sided p < 0.05 is considered significant. No correction for multiple comparisons applied. Abbreviations: SD standard deviation, MVPA moderate-vigorous physical activity.

**Table 2 | Sample size and median follow-up period in each lifestyle in the pooled analytic sample (N = 32,033)**

| | Less than weekly social contact | Weekly social contact |
|---|---|---|
| **Smoking** | | |
| *Heavy alcohol consumption* | | |
| Less than weekly MVPA | 748, 7.5 (0–12) | 589, 8 (1–13) |
| Weekly MVPA | 731, 8 (0–12) | 556, 10 (5–13) |
| *No-to-moderate alcohol consumption* | | |
| Less than weekly MVPA | 1177, 6 (0–11) | 666, 8 (1–12) |
| Weekly MVPA | 876, 8 (0–12) | 557 10 (4–12) |
| **Non-smoking** | | |
| *Heavy alcohol consumption* | | |
| Less than weekly MVPA | 1783, 8 (0–12) | 2403, 10 (4–13) |
| Weekly MVPA | 1702, 10 (0–12) | 2589, 10 (5–13) |
| *No-to-moderate alcohol consumption* | | |
| Less than weekly MVPA | 6365, 6 (0–11) | 4106, 8 (2–13) |
| Weekly MVPA | 3520, 8 (0–12) | 3665, 10 (4–13) |

N, median follow-up period in years (interquartile range) shown. Abbreviations: MVPA moderate and vigorous physical activity.

**Table 3 | Memory decline over 10 years in each lifestyle (N = 32,033)**

| | Less than weekly social contact | Weekly social contact |
|---|---|---|
| **Smoking** | | |
| *Heavy alcohol consumption* | | |
| Less than weekly MVPA | −0.33 (−0.42 to −0.24) | −0.29 (−0.38 to −0.20) |
| Weekly MVPA | −0.37 (−0.46 to −0.28) | −0.27 (−0.37 to −0.18) |
| *No-to-moderate alcohol consumption* | | |
| Less than weekly MVPA | −0.30 (−0.37 to −0.22) | −0.31 (−0.40 to −0.22) |
| Weekly MVPA | −0.33 (−0.41 to −0.25) | −0.20 (−0.29 to −0.10) |
| **Non-smoking** | | |
| *Heavy alcohol consumption* | | |
| Less than weekly MVPA | −0.24 (−0.29 to −0.18) | −0.27 (−0.31 to −0.23) |
| Weekly MVPA | −0.21 (−0.26 to −0.15) | −0.23 (−0.27 to −0.19) |
| *No-to-moderate alcohol consumption* | | |
| Less than weekly MVPA | −0.20 (−0.23 to −0.17) | −0.20 (−0.24 to −0.17) |
| Weekly MVPA | −0.23 (−0.27 to −0.19) | −0.20 (−0.23 to −0.16) |

Memory decline over 10 years (95% confidence interval) from age 65 years in standard deviations. Based on models adjusted for age at baseline, gender, country, education, wealth, chronic conditions. Abbreviations: MVPA moderate and vigorous physical activity.

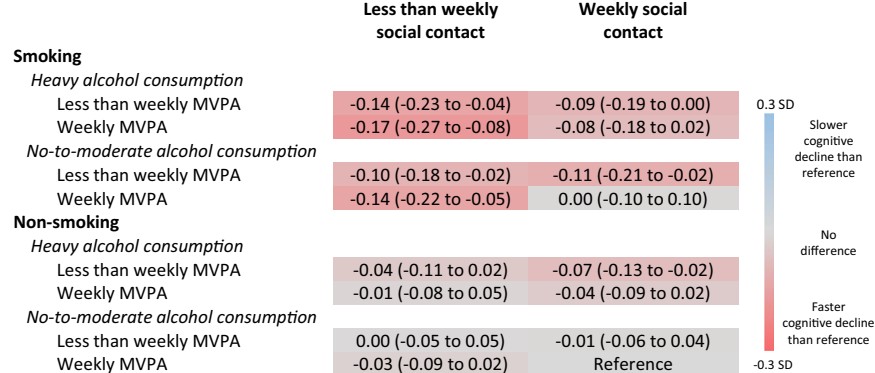

**Fig. 1 | Difference in memory decline over 10 years between the recommendation compliant lifestyle and other lifestyles (N = 32,033).** Difference in memory decline over 10 years (95% confidence interval) in standard deviations between recommendation compliant lifestyle (reference) and other lifestyles. Negative value indicates memory decline is slower for the reference lifestyle. Based on models adjusted for age at baseline, gender, country, education, wealth, chronic conditions. Abbreviations: MVPA moderate and vigorous physical activity, SD standard deviations.

weekly MVPA, and weekly social contact declined −0.09 SD (−0.19 to 0.00) more over 10 years than the reference lifestyle, and those with weekly MVPA declined −0.08 (−0.18 to 0.02) more; these differences did not reach statistical significance ($p = 0.062$ and $p = 0.117$ respectively). Participants who reported smoking, no-to-moderate alcohol consumption, less-than-weekly MVPA, and weekly social contact declined −0.11 (−0.21 to −0.02) more over 10 years than the reference lifestyle ($p = 0.020$). Participants who reported smoking, no-to-moderate alcohol consumption, weekly MVPA, and weekly social contact had no difference in memory decline compared with the reference lifestyle [difference in decline over 10 years=0.00 [−0.10 to 0.10]; $p = 0.955$).

All but one of the non-smoking lifestyles had similar memory decline to the reference lifestyle. The exception was those who reported non-smoking, heavy alcohol consumption, less-than-weekly MVPA, and weekly social contact declined −0.07 SD (−0.13 to −0.02) more over 10 years than the reference lifestyle ($p = 0.009$).

Fluency decline over 10 years from age 65 for each lifestyle is shown in Table 4. After adjustment for covariates, participants reporting the reference lifestyle had fluency scores that declined −0.19 SD (−0.22 to −0.15) during 10 years of follow-up.

The difference in fluency decline over 10 years between the reference lifestyle and the other lifestyles is shown in Fig. 2. There were generally small or negligible differences in fluency decline between the reference lifestyle and other lifestyles. Three lifestyles were exceptions: participants who reported smoking, heavy alcohol consumption, less-than-weekly MVPA, and less-than-weekly social contact declined −0.16 SD (−0.25 to −0.07) more than the reference lifestyle during the 10-year follow-up period ($p < 0.001$), those who reported smoking, heavy alcohol consumption, weekly MVPA, and less-than-weekly social contact declined −0.11 (−0.20 to −0.02) more ($p = 0.019$), and participants who did not smoke and reported no-to-moderate alcohol consumption, less-than-weekly MVPA, and less-than-weekly social contact declined 0.11 (0.07 to 0.16) less over 10 years than the reference lifestyle ($p = 0.044$).

**Additional analyses**

Results remained consistent after excluding participants who reported abstaining from alcohol use (Table S7, S8). Overall, behaviors remained consistent during the 15-year follow up period where 0.9% of participants reported a change in smoking status, 4.4% of participants reported a change in MVPA group, and 2.6% of participants reported a

change in social contact group. Adjustments for practice effect (Tables S9, S10), body mass index (Table S11, S12), or hearing difficulty (Table S13, S14) did not change the results. There were also negligible differences in associations between smoking and cognitive decline when current smokers were compared with former smokers or never-smokers (Table S6).

## Discussion

The key finding arising from this examination of healthy lifestyle and 10-year cognitive decline is that associations between lifestyle and cognitive decline primarily depended on whether participants reported smoking. For those who reported current smoking, cognitive decline was generally faster than the reference lifestyle, except for those with recommendation-compliant alcohol, MVPA, and social contact habits. By contrast, cognitive decline was generally similar between all non-smoking lifestyles, regardless of alcohol, physical activity, or social contact habits. Taken together, the results suggest an important role of smoking habits in shaping cognitive ageing.

A main strength of this study is its large study population and long follow-up period, comprising over 32,000 participants from 14 European countries followed up over a period of up to 15 years. This

enabled us to investigate 16 lifestyle profiles with sufficient power to detect clinically relevant differences in cognitive decline, where previous studies are limited in the number of lifestyles considered or use healthy lifestyle indices. As prodromal dementia might influence behavior years before diagnosis[22], we reduced the likelihood of reverse causation driving the results we observed by focussing on cognitive decline rather than cross-sectional cognitive performance and by excluding participants with suspected cognitive impairment or who reported dementia diagnosis. Finally, we mitigated differences between countries and cohorts by standardizing cognitive scores to each country, adjusting for the country, and evaluating including random effects on cohort and country.

There are limitations to the present study. First, all behaviors were self-reported; nonetheless, self-reported measures show associations with a range of health outcomes, including dementia[2], suggesting they have clinical relevance and evidence for behavior guidelines is based on self-report[23]. We were not able to account for behavior changes during the follow-up period due to the lack of availability of alcohol variables at later waves of follow-up; however, when we examined smoking, social contact, and MVPA, these habits remained mostly consistent during the follow-up period in our analytic sample, suggesting changes in behavior are less likely to have influenced the results. Due to a lack of data, we could not account for factors such as medication use, which might confound associations between lifestyle and cognitive function. As such, residual confounding might limit causal inference. The inclusion of four behavioral factors may also limit causal inference; notably, we were unable to include sleep or diet though both behaviors may be relevant to cognitive ageing[24,25], as these data were not available in SHARE. Consideration of body mass index might partially account for dietary choices; the results were nonetheless unchanged after taking body mass index into account. Differential attrition could have impacted results, as participants with less healthy lifestyles may be more likely to drop out of the cohort; however, follow-up durations were generally similar for those reporting less healthy and healthier behaviors, with differences in follow-up duration corresponding to one wave, suggesting the impact on results is likely to be minor overall. Future research should examine the generalizability of the results in non-European study populations, reproduce analyses using objective measures of physical activity and clinically derived measures of chronic conditions, include more behaviors, examine other cognitive domains and dementia, and effect modification by APOE status.

Our results are in accordance with a large body of evidence that finds associations of non-smoking and low-to-moderate alcohol consumption with better cognitive function[26]. While short-term exposure

## Table 4 | Fluency decline over 10 years in each lifestyle (N = 32,033)

| | Less than weekly social contact | Weekly social contact |
|---|---|---|
| **Smoking** | | |
| *Heavy alcohol consumption* | | |
| Less than weekly MVPA | −0.35 (−0.43 to −0.26) | −0.17 (−0.26 to −0.09) |
| Weekly MVPA | −0.29 (−0.38 to −0.21) | −0.21 (−0.29 to −0.12) |
| *No-to-moderate alcohol consumption* | | |
| Less than weekly MVPA | −0.23 (−0.30 to −0.16) | −0.15 (−0.23 to −0.06) |
| Weekly MVPA | −0.21 (−0.29 to −0.14) | −0.26 (−0.34 to −0.17) |
| **Non-smoking** | | |
| *Heavy alcohol consumption* | | |
| Less than weekly MVPA | −0.18 (−0.24 to −0.13) | −0.18 (−0.21 to −0.14) |
| Weekly MVPA | −0.18 (−0.23 to −0.12) | −0.13 (−0.17 to −0.10) |
| *No-to-moderate alcohol consumption* | | |
| Less than weekly MVPA | −0.07 (−0.10 to −0.04) | −0.15 (−0.18 to −0.12) |
| Weekly MVPA | −0.15 (−0.19 to −0.12) | −0.19 (−0.22 to −0.15) |

Fluency decline over 10 years (95% confidence interval) from age 65 years in standard deviations. Based on models adjusted for age at baseline, gender, country, education, wealth, and chronic conditions. Abbreviations: MVPA moderate and vigorous physical activity.

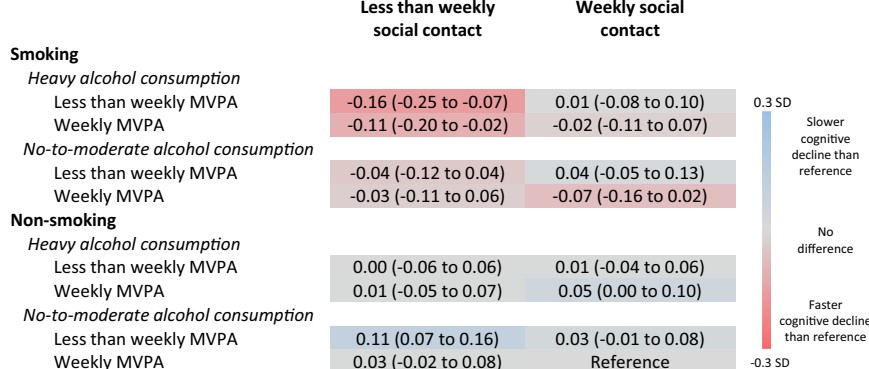

**Fig. 2 | Difference in fluency decline over 10 years between the recommendation compliant lifestyle and other lifestyles (N = 32,033).** Difference in fluency decline over 10 years (95% confidence interval) in standard deviations between the recommendation compliant lifestyle (reference) and other lifestyles. Negative value indicates fluency decline is slower for the reference lifestyle. Based on models adjusted for age at baseline, gender, country, education, wealth, and chronic conditions. Abbreviations: MVPA moderate and vigorous physical activity, SD standard deviations.

to nicotine may be beneficial for cognitive function, smoking might contribute to cognitive decline due to its negative impacts on the cardiovascular system[27]. Mechanisms underlying associations of light-to-moderate alcohol consumption with cognitive function are not well understood, although light-to-moderate alcohol consumption has been associated with reduced activity in stress-associated brain regions[28]. While previous evidence suggests complete abstinence from alcohol consumption is associated with worse cognitive performance compared with light-to-moderate consumption[29], those who abstain from alcohol use commonly do so due to poor health[30], suggesting this result may be due in part to incomplete adjustment for confounding. Our sensitivity analysis excluding participants who abstained from alcohol use suggested abstainers did not substantively impact the results we observed.

Physical activity is thought to be protective against cognitive decline and dementia in part due to its ability to function as a buffer against aging-related neuronal loss and strengthen compensatory mechanisms employed against aging-related neurodegeneration[31]; however, there is also evidence that associations between physical activity and dementia may be due to the effects of preclinical or pro-dromal dementia on physical activity rather than a causal effect of physical activity on dementia[32]. This is consistent with our finding that recommendation-compliant physical activity was not associated with a slower rate of cognitive decline in this cognitively healthy study population, although this result also could have been due to over-reporting of physical activity.

Finally, social activity and engagement may promote neuroprotection by moderating associations between brain atrophy and cognitive function; social bonding might also moderate the association between perceived stress and cognitive function[33]. While we did not find an independent association between social contact and cognitive decline, when considered with other behaviors, social contact appeared to contribute to reductions in cognitive decline.

Previous studies have found that when healthy behaviors are combined into a healthy lifestyle index, more healthy behaviors are associated with better cognitive performance or slower cognitive decline[5]. This body of evidence suggests the least healthy lifestyle should be associated with the fastest cognitive decline, with each progressively healthier lifestyle associated with a stepwise decrease in the rate of decline. Instead, we found that whether engaging in more healthy behaviors was associated with a decrease in the rate of cognitive decline primarily depended on participants' smoking habits. These results suggest certain combinations of behaviors may be particularly important for the maintenance of cognitive function.

A reduction in cognitive decline of 40–50% in a study population in the preclinical period of Alzheimer's disease has been found to correspond to a three-year delay in clinically relevant cognitive impairment[34]. Addressing modifiable risk factors to reduce the rate of cognitive decline can lead to improvements in quality of life and delayed onset of cognitive impairment and dementia. This analysis puts forth key combinations of behaviors associated with the rate of cognitive decline in a population not yet experiencing pathological cognitive symptoms. The findings suggest that not smoking may be sufficient to attenuate differences in the rate of cognitive decline attributable to lifestyle; however, where smoking cessation is not feasible, emphasis on other healthy habits may, to an extent, compensate for smoking habits. It is also necessary to emphasize that while in the present study, differences in memory and fluency decline appeared to depend primarily on smoking status, evidence suggests alcohol, physical activity, and social contact habits nonetheless contribute to a range of other health outcomes, including dementia[3].

There is robust evidence that smoking, alcohol consumption, physical activity, and social contact are associated with cognitive health, both when examined independently and when combined into a healthy lifestyle index[26]. The present results suggest examining behaviors independently or in indices may not fully capture associations between lifestyle and cognitive decline and point to specific combinations of behaviors that may contribute to the maintenance of cognitive function. This study highlights the importance of a holistic approach to lifestyle interventions that considers an individual's existing habits to maximize long-term cognitive benefits.

## Methods
### Data sources
Data were drawn from the English Longitudinal Study of Ageing (ELSA) and the Survey of Health, Ageing and Retirement in Europe (SHARE). ELSA is a nationally representative cohort study of the English population aged ≥50 years. Data collection began in 2002/03 with follow-up every two years thereafter. SHARE is a nationally representative cohort study of residents aged ≥50 years in 28 European countries and Israel, with follow-up in 2004/05, 2006/07, 2010/11, 2013, 2015, 2017, and 2019/20. These cohorts have a similar survey design and implementation to facilitate cross-country comparisons and harmonization; details are available elsewhere[35,36]. Both cohorts were granted relevant local ethics approval, with written informed consent given at each interview.

The present analysis included waves 2-8 of ELSA (2004/06-2018/19) and waves 2 and 4-8 (2006/07-2019/20) of SHARE, comprising up to 15 years of follow-up. Waves 2 and 4-8 of SHARE included 13 of the 28 European countries. Wave 3 of SHARE is a life history module that is not part of the main survey. Respondents aged ≥50 years from European countries participating in wave 2 of SHARE or ELSA with at least one round of cognitive testing were eligible for inclusion in the analyzes. ELSA and SHARE data were pooled for analysis.

### Lifestyle and behaviors
The four behaviors considered were smoking, alcohol consumption, physical activity, and social contact, which were assessed at the baseline wave of the present study (wave 2 in both SHARE and ELSA) and were not updated during follow-up.

To assess smoking, participants were asked to report whether they currently consider themselves a smoker as well as whether they had smoked previously. Alcohol consumption was assessed by asking participants about their average alcohol consumption in the past six months (SHARE) or maximum alcohol consumption in the past seven days (ELSA). Participants reported the number of drinks they consumed, with examples of what volume constituted one drink for each type of alcoholic beverage provided. The number of alcoholic beverages SHARE participants could report consuming daily was capped at 70; SHARE participants reporting 70 drinks per day were excluded from analyzes. To assess physical activity, ELSA and SHARE participants reported the current frequency of participation in moderate and vigorous physical activity (rarely/never, 1–3 times monthly, weekly, or greater than weekly), with examples given for each physical activity intensity category. This measure of physical activity has been found to be moderately correlated with accelerometer-assessed physical activity in a subset of ELSA participants[37]; WHO guidelines of 150 min per week of moderate or 75 min per week of vigorous physical activity are also based on self-report measures[23]. Finally, to examine social contact, participants were asked whether they currently had at least weekly social contact with relatives and/or friends or whether they participated in weekly social activities.

To facilitate interpretation of the results, behaviors were dichotomized into categories corresponding to more recommendation compliant and less recommendation compliant behavior based on WHO and United States guidelines[26,38]. These dichotomized behaviors made up an individual's lifestyle, where there were 16 distinct lifestyles corresponding to all possible combinations of the four behaviors.

Smoking was categorized into a current smoker (smoking) and not a current smoker (non-smoking). The dichotomization of alcohol

consumption was based on current United States guidelines[38]. Alcohol consumption was categorized into no-to-moderate consumption (up to 2 drinks per day for men or 1 drink per day for women) or heavy consumption. Physical activity was categorized into those who reported participating in both moderate and vigorous physical activity at least weekly (moderate-plus-vigorous physical activity or weekly MVPA) compared with those who reported less than weekly moderate or vigorous physical activity (less-than-weekly MVPA). We chose to require both weekly moderate and vigorous activity because all categorizations of MVPA requiring less activity resulted in 75–85% of the sample being categorized into the weekly MVPA group, yielding insufficient sample sizes to examine the cognitive decline in lifestyles with less-than-weekly MVPA. Finally, social contact was categorized into those who reported weekly social contact (weekly social contact) compared with those who did not (less-than-weekly social contact). The recommendation-compliant lifestyle refers to participants reporting non-smoking, no-to-moderate alcohol consumption, weekly MVPA, and weekly social contact.

## Cognitive function

The cognitive domains examined were episodic memory and verbal fluency. These cognitive domains show decline with ageing[39], and dementia[40], making them appropriate cognitive domains to examine risk factors for cognitive decline. Tests of episodic memory and verbal fluency were also the same in both cohorts and administered at enough waves to facilitate the examination of long-term cognitive trajectories.

Episodic memory was assessed using the Consortium to Establish a Registry for Alzheimer's Disease immediate and delayed recall tasks[41]. Participants were given a 10-word list and asked to recall it immediately and after a delay, with scores on both tests summed to yield an overall recall score. Verbal fluency was assessed using the animal naming task[42], in which participants were required to name as many animals as possible within a one-minute period. Memory tasks were administered at every wave; animal naming was not administered during ELSA wave 6. Given absolute differences in cognitive scores between countries, cognitive scores were standardized by country using the mean and standard deviation of each country's scores at baseline.

To reduce the impact of prodromal dementia on behavior, participants reporting having received a dementia diagnosis at any interview during the follow-up period or who had cognitive scores suggesting cognitive impairment were excluded from analyzes. A cognitive score in either cognitive domain more than 1.5 standard deviations below the mean for an individual's 5-year age group in their country was considered to be evidence of potential cognitive impairment; this cut-off is widely used to identify cognitive impairment when clinical evaluation is not possible[43]. We excluded from analyses participants whose scores suggested cognitive impairment for at least two waves of follow-up. We did not exclude participants with just one wave of cognitive impairment unless it was their only wave of participation; this was done to retain participants experiencing transient cognitive impairment as up to 55% of those diagnosed with mild cognitive impairment may revert to normal cognitive function[44].

## Covariates

Covariates were selected on the basis of previous evidence of associations with lifestyle and cognitive function and were ascertained by self-report. Sociodemographic covariates included gender (man or woman), age at baseline in years, and country (Table S15). Socioeconomic covariates included education (less than upper secondary, upper secondary, or tertiary and above; categorized based on the International Standard Classification of Education 2011[45]) and household non-housing wealth. Wealth was standardized to each country by year and converted into quintiles, with the highest quintile corresponding to the greatest wealth. Chronic conditions were ascertained

based on self-report of clinical diagnosis and included high blood pressure, diabetes, cardiovascular conditions (including heart disease and stroke), cancer, lung disease, high cholesterol, and psychiatric conditions. Self-report of clinical diagnosis of chronic conditions has been shown to have good agreement with ascertainment based on medical records[46]. All covariates except gender and education—which were drawn from the baseline wave—were time-varying and were recorded at each interview or imputed from the closest wave if missing.

## Statistical analysis

After imputation of time-varying covariates, participants who were still missing data were dropped from analyzes with one exception: we singly imputed behaviors for 662 participants (2.1% of the analytic sample) missing one behavior out of the four to retain participants who otherwise had a full set of covariates and cognitive scores. This imputation was based on logistic regression models that included country, gender, marital status (married/partnered or not), education level, labor force status (retired, employed, unemployed), age at baseline, all other behaviors, chronic conditions, memory and fluency scores, and body mass index.

We then described participant characteristics in the pooled analytic sample at baseline for each of the four behaviors, with Pearson's $x^2$ test used to examine associations with categorical covariates and $t$-test with continuous covariates.

Linear mixed models were used to examine associations between lifestyle and 10-year memory and fluency decline. Linear mixed models use all available data regardless of length of follow-up, handle non-monotone missingness patterns, and data missing-at-random[47]. These models included a random intercept and slope at the individual level with an unstructured covariance matrix to account for the correlation of repeated measurements on each participant and used time since baseline in years as the timescale. We also fitted models that included an additional random intercept on cohort or country but found that these terms explained negligible variation in cognitive trajectory and, therefore, omitted them in the reported results to simplify the model.

We first fitted models adjusted for sociodemographic and socioeconomic covariates and chronic conditions to examine independent associations between each of the behaviors (smoking, alcohol consumption, MVPA, and social contact) and memory and fluency decline over 10 years. These models included *time* (years since baseline), age at baseline in years (denoted $age_0$), the interaction between *time* and $age_0$ (denoted *time x $age_0$*), all covariates, each behavior, and the interaction between each behavior and *time* (denoted *behavior x time*). $Age_0$ was centered at 65 years, the mean in the pooled analytic sample. After determining that memory had a non-linear association with time, we also included $time^2$ and *behavior x $time^2$* terms in the memory model only.

We then fitted similar models to examine differences in memory and fluency decline over 10 years between the recommendation-compliant lifestyle (non-smoking, no-to-moderate alcohol consumption, weekly MVPA, and weekly social contact) and the other lifestyles. In these models, instead of *behavior* and *behavior x time* terms, we included *lifestyle*, and a *lifestyle x time* interaction term, with the recommendation-compliant lifestyle as the reference lifestyle. Examination of *lifestyle x time x gender* and *lifestyle x time x $age_0$* terms suggested associations between lifestyle and cognitive decline were similar between men and women and for different ages; as such, we performed analyses in the entire analytic sample without stratifying by gender or age at baseline.

We first used these models to estimate how much memory and fluency scores declined over 10 years from age 65 in each lifestyle; this was done to contextualize differences in cognitive decline between lifestyles. We then determined the difference in memory and fluency decline over 10 years between the reference and the other lifestyles. All

analyzes were performed in StataMP 18.0 with a two-sided *p* <0.05 considered significant.

## Additional analyses

As individuals who abstain from alcohol use frequently do so due to poor health[30], and as a result abstaining from alcohol consumption has been associated with adverse health outcomes including poor cognitive performance[29], we examined how excluding abstainers from analyses impacted the results. As we could not account for changes in behavior during the follow-up period due to the lack of availability of consistent alcohol variables, we also examined the stability of the other three behaviors during the follow-up period to determine whether changes in behavior were likely to have influenced the results.

In further analyses, we examined the impact of adjusting for practice effect, body mass index, and self-reported hearing difficulty on the results as well as independent associations between smoking and cognitive decline when smoking was split into three categories (never smokers, former smokers, and current smokers).

## Reporting summary

Further information on research design is available in the Nature Portfolio Reporting Summary linked to this article.

## Data availability

ELSA and SHARE data are freely available to researchers. Access can be obtained after registration with the UK data service (https://www.elsa-project.ac.uk/accessing-elsa-data) or with the SHARE project (https://share-eric.eu/data/data-access).

## Code availability

Example code, a simulated dataset, and instructions for their use are available[48].

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

## Acknowledgements

ELSA is funded by the National Institute on Aging (RO1AG017644) and by UK Government Departments coordinated by the National Institute for Health and Care Research (NIHR). SHARE has received funding from the European Union under grant agreements No 101102412 and the European Union's Horizon 2020 research and innovation program under grant agreements No 870628 and No 101015924. MB and LB are supported by the Economic and Social Research Council (ES/T014091/1). The funders had no role in study design, data collection and analysis, decision to publish, or preparation of the manuscript.

## Author contributions

Conceptualization: M.B., and A.S. Methodology: M.B., G.M.T., and A.S. Validation: G.M.T., L.B., and A.S. Formal analysis: M.B. Data curation: M.B., and A.S. Writing–original draft preparation: M.B. Writing–review and editing: All authors. Visualization: M.B. Supervision: A.S. Funding acquisition: A.S.

## Competing interests

The authors declare no competing interests.
