## [Peer Review File · Nature Communications]

Reviewers' Comments:

Reviewer #1:

Remarks to the Author:

Mikaela Bloomberg et al.'s longitudinal study aims to evaluate the impact of 16 lifestyles, comprising four behavioral factors (smoking, alcohol consumption, physical activity, and social contact) on memory and fluency trajectories among 32,033 participants aged over 50 years from ELSA and SHARE cohorts. Evaluating up to 15 years of data, The analysis reveals smoking's pivotal role within combined lifestyles, suggesting that not smoking may be sufficient to substantially attenuate differences in cognitive decline attributable to lifestyle. Notably, the study emphasizes that the specific combination of healthy behaviors is more important than the number of healthy behaviors. These findings will provide valuable guidance for clinicians or policymakers in promoting cognitive health among adults. However, certain aspects necessitate clarification.

Methods:

1. Clarification is required regarding the inclusion/exclusion criteria. While table S1 excludes participants with dementia or cognitive impairment, it remains unclear whether this exclusion pertains to baseline or the entire follow-up period. Additionally, how were individuals with conditions like hearing loss, aphasia, stroke, or other factors impacting cognitive assessment or physical activity at baseline treated?
2. More detail is needed on the self-reporting timeframe for healthy behaviors. Specifically, what period was considered for reporting these behaviors (e.g., last month, one year, two years), especially for smoking status?
3. The approach to handling missing covariates warrants clarification. Table S1 suggests exclusion, yet page 9 of the manuscript implies imputation. What was the strategy for managing missing data?
4. Were there baseline differences in cognitive function among various lifestyle profiles? Accounting for baseline cognitive function is crucial to mitigate confounding effects.
5. The practice effect of cognitive assessment should be considered as a covariate in the statistical analysis.
6. Did the authors consider the potential effects of some medications for chronic diseases on cognitive function?
7. Details regarding the number of participants lost or deceased during the 15-year follow-up and any differential effects across lifestyles are essential. Further insight into how these cases were handled in the analysis is needed.
8. How was dementia diagnosed, and were the diagnostic criteria were consistent across both cohorts?

Results and discussion:

1. Participants who did not smoke or had no-to-moderate drinking habits been reported to have more chronic diseases. The study should address how this result was factored into the analysis.
2. Given the last follow-up was in 2019/2020, it would be relevant to discuss the potential influence of the COVID-19 pandemic on the cognitive function of older adults in the present study.
3. The author concluded that associations between lifestyle and cognitive decline primarily depended on whether participants reported smoking, and examining behaviours independently or in indices does not fully capture associations between lifestyle and cognitive decline. It would be valuable for the authors to provide independent analysis results for the four health behaviors and discuss their relevance alongside the current findings.

Reviewer #2:

Remarks to the Author:

This paper is clear and well-written, and presents an original analysis of the association between different lifestyles and cognitive decline.

I do have some reservations about the methods, the depth of the analyses, and conclusions drawn, that I think should lead to some revisions.

My main queries are:

1. In the abstract and elsewhere, non-to moderate alcohol intake is referred to as healthier. But this is not borne out by the results shown in Table 1, and therefore 'healthier' is misleading.
2. Smoking is the dominant factor, as shown by the authors. I think that once this has been

shown, then more analyses are required to explore this further. In particular, the reference group of non-smokers will comprise past smokers (including recent ones) and never smokers. These are not likely to fit the same risk profile. I suggest that the smoking variable should be in 3 categories: never, past and current.

3. Addressing reverse causation. The authors have tried to minimise bias due to reverse causation but I do not think they have gone far enough. In particular, updating the exposure at every wave will introduce reverse causation bias, since some of those who change their behaviour might be caused to do so by incipient cognitive changes. I suggest that a further analysis is required which uses baseline exposures and does not update them. It is also not possible to say that reverse causation will not be affecting the results, as the follow-up time was about a decade. The best you can hope for, is that it is minimised.

4. Imputation of missings. I have some doubt over whether it also introduces bias to impute missings using the outcomes as well as the exposure. In any case, this could be avoided if you use baseline exposure.

5. Reporting non-significant differences between groups. I think the results should be revised, to make it clearer which differences are actually statistically significantly different. When looking at the confidence intervals that overlap in the tables, I would not think that many of the subgroups are significantly different from each other – except the smoking.

6. Omission of BMI. The rationale given for the omission of BMI was not convincing. There is strong evidence that BMI is a risk factor for dementia (see studies by Kivimaki et al and by the Million Women Study team) after the removal of early years of follow-up. BMI has a stronger association with dementia than physical activity. In addition, it is not a strong argument to say that BMI is on the causal pathway from physical activity to cognitive decline, since BMI is much more strongly influenced by energy intake than by physical activity.

7. Discussion and interpretation of the results.

line 348, line 425 and concluding paragraph – I disagree that there is a complex role for these factors. The results presented suggest that it is simple, in that only smoking really makes a big difference to cognitive performance. Therefore efforts should be made to encourage cessation and efforts do not need to be directed at the other factors. I do not see that there has been clear evidence presented here that the other factors investigated here could compensate.

line 402 and line 430 – There is evidence that the reported association between physical activity and dementia is biased by reverse causation (see studies by the Whitehall II group and by the Million Women Study group).

Reviewer #1

1. *Mikaela Bloomberg et al.'s longitudinal study aims to evaluate the impact of 16 lifestyles, comprising four behavioral factors (smoking, alcohol consumption, physical activity, and social contact) on memory and fluency trajectories among 32,033 participants aged over 50 years from ELSA and SHARE cohorts. Evaluating up to 15 years of data, The analysis reveals smoking's pivotal role within combined lifestyles, suggesting that not smoking may be sufficient to substantially attenuate differences in cognitive decline attributable to lifestyle. Notably, the study emphasizes that the specific combination of healthy behaviors is more important than the number of healthy behaviors. These findings will provide valuable guidance for clinicians or policymakers in promoting cognitive health among adults. However, certain aspects necessitate clarification.*

OUR RESPONSE: Thank you for the positive feedback. We hope that the following revisions will provide the necessary clarification.

2. *Clarification is required regarding the inclusion/exclusion criteria. While table S1 excludes participants with dementia or cognitive impairment, it remains unclear whether this exclusion pertains to baseline or the entire follow-up period. Additionally, how were individuals with conditions like hearing loss, aphasia, stroke, or other factors impacting cognitive assessment or physical activity at baseline treated?*

OUR RESPONSE: Participants reporting dementia diagnosis at any time during the follow-up period or who had cognitive scores suggesting cognitive impairment for two or more waves of participation were excluded from analyses completely. These participants were excluded because prodromal dementia symptoms can precede dementia diagnosis by many years and may impact behaviour.¹ Because dementia ascertainment is based on self-report of physician diagnosis, dementia may be under-reported in the analytic sample, which is why we also included evidence of sustained cognitive impairment as grounds for exclusion. We have revised the Methods section to indicate that dementia diagnosis or evidence of cognitive impairment at any time during the follow-up period resulted in exclusion from analyses:

Methods, page 18, paragraph 3: "To reduce the impact of prodromal dementia on behaviour, participants reporting having received a dementia diagnosis at any interview during the follow-up period or who had cognitive scores suggesting cognitive impairment were excluded from analyses. [...] We excluded from analyses participants whose scores suggested cognitive impairment for at least two waves of follow-up. We did not exclude participants with just one wave of cognitive impairment unless it was their only wave of participation; this was done to retain participants experiencing transient cognitive impairment as up to 55% of those diagnosed with mild cognitive impairment may revert to normal cognitive function.⁴⁵"

We accounted for stroke (and thus stroke-induced conditions such as aphasia) by adjusting for diagnosis of stroke, which we have now clarified in the Methods section:

Methods, page 19, paragraph 2: "Covariates were selected on the basis of previous evidence of associations with lifestyle and cognitive function and were ascertained by self-report. [...] Chronic conditions were ascertained based on self-report of clinical diagnosis and included high blood

pressure, diabetes, cardiovascular conditions (including heart disease and stroke), cancer, lung disease, high cholesterol, and psychiatric conditions.”

Methods, page 20, paragraph 4: “We first fitted models adjusted for sociodemographic and socioeconomic covariates and chronic conditions to examine independent associations between each of the behaviours (smoking, alcohol consumption, MVPA, and social contact) and memory and fluency decline over 10 years. [...] We then fitted similar models to examine differences in memory and fluency decline over 10 years between the recommendation compliant lifestyle (non-smoking, no-to-moderate alcohol consumption, weekly MVPA, and weekly social contact) and the other lifestyles.”

We thank the reviewer for pointing out the potential confounding effect of hearing difficulties on the association between physical activity and cognitive function. We have now added a sensitivity analysis to the Supplemental materials (Tables S13-S14) where we adjusted for participant-reported hearing difficulty. The results were unchanged. We have revised the manuscript as follows:

Methods, page 22, paragraph 2: “In further analyses, we examined the impact of adjusting for practice effect, body mass index, and self-reported hearing difficulty on the results [...]”

Results, page 10, paragraph 1: “Adjustment for practice effect (Tables S9-S10), body mass index (Tables S11-S12), or hearing difficulty (Tables S13-S14) did not change the results.”

References

1. Ramakers IH, Visser PJ, Aalten P, et al. Symptoms of preclinical dementia in general practice up to five years before dementia diagnosis. *Dement Geriatr Cogn Disord* 2007; **24**(4): 300-6.

3. *More detail is needed on the self-reporting timeframe for healthy behaviors. Specifically, what period was considered for reporting these behaviors (e.g., last month, one year, two years), especially for smoking status?*

OUR RESPONSE: We have clarified the timeframe for reporting healthy behaviours in the Methods section as suggested:

Methods, page 16, paragraph 1: “To assess smoking, participants were asked to report whether they currently consider themselves a smoker as well as whether they had smoked previously. Alcohol consumption was assessed by asking participants about their average alcohol consumption in the past six months (SHARE) or maximum alcohol consumption in the past seven days (ELSA). [...] To assess physical activity, ELSA and SHARE participants reported current frequency of participation in moderate and vigorous physical activity (rarely/never, 1-3 times monthly, weekly, or greater than weekly), with examples given for each physical activity intensity category.”

4. *The approach to handling missing covariates warrants clarification. Table S1 suggests exclusion, yet page 9 of the manuscript implies imputation. What was the strategy for managing missing data?*

OUR RESPONSE: Time-varying covariates were imputed from the nearest wave if missing. As covariate missingness after performing this imputation was very minor (<1% in both SHARE and ELSA), participants still missing covariates after imputation were dropped from the analysis. We also

imputed data for individuals who were missing a single behaviour out of the four behaviours to retain individuals who had full data for covariates and cognitive performance but were just missing a single behaviour. We have clarified the Methods and Results sections to reflect this:

Methods, page 19, paragraph 3: “After imputation of time-varying covariates, participants who were still missing data were dropped from analyses with one exception: we singly imputed behaviours for 662 participants (2.1% of the analytic sample) missing one behaviour out of the four to retain participants who otherwise had a full set of covariates and cognitive scores.”

Results, page 5, paragraph 1: “Of 9,131 respondents aged 50 years and older participating in wave 2 of ELSA, 1,088 (11.9%) were missing more than one behaviour at baseline and therefore did not have missing behaviour data imputed, 1,821 (19.9%) either reported dementia diagnosis or had cognitive scores suggesting cognitive impairment, 5 (<0.1%) were missing cognitive scores at all waves of follow-up, and 24 (0.3%) were missing covariates; these respondents were excluded leading to 6,193 ELSA respondents being included in analyses (Figure S1).”

Participants missing more than one behaviour were dropped from analyses. While this might limit the generalisability of the study, as the study is longitudinal, it does not induce selection bias as it would with a cross-sectional study. We have highlighted the necessity of research in other study populations to examine the generalisability of the results in the Limitations section:

Discussion, page 12, paragraph 1: “Future research should examine the generalisability of the results in non-European study populations [...]”

5. *Were there baseline differences in cognitive function among various lifestyle profiles?
Accounting for baseline cognitive function is crucial to mitigate confounding effects.*

OUR RESPONSE: While we agree with the reviewer that adjusting for baseline conditions is important in some analyses, we disagree that it is appropriate to adjust for baseline cognitive score in the present study. The methodological approach employed, linear mixed effects models, is such that population average and individual-specific baseline cognitive function and rate of change are estimated, and therefore, it would not be correct to adjust for baseline cognitive function. Furthermore, in circumstances where exposures are associated with baseline health status and health status may precede baseline assessment—as in the present study—adjustment for baseline cognitive performance can induce a spurious statistical association between the exposure and change in cognitive score.² Under these circumstances, the unbiased causal estimate is produced without baseline adjustment. As such, we did not adjust for baseline cognitive score in our models. However, we agree that including baseline differences in cognitive performance could provide additional context. We have revised the Supplemental materials to include cognitive performance at baseline in each lifestyle (Tables S4-S5). While there were differences in baseline cognitive performance between lifestyles, these differences also reflect the heterogeneity of age distributions between lifestyles. We have revised the Results section to refer to baseline differences in cognitive performance as follows:

Results, page 7, paragraph 1: “Mean baseline cognitive performance differed between lifestyles, ranging from -0.33 SD (SD=1.04) for memory or -0.32 SD (0.99) for fluency in the non-smoking, no-to-moderate alcohol consumption, less-than-weekly MVPA, less-than-weekly social contact

lifestyle to 0.33 SD (0.94) for memory or 0.32 SD (1.04) for fluency in the non-smoking, heavy alcohol consumption, weekly MVPA, weekly social contact lifestyle (Tables S4-S5).”

References

2. Glymour MM, Weuve J, Berkman LF, Kawachi I, Robins JM. When Is Baseline Adjustment Useful in Analyses of Change? An Example with Education and Cognitive Change. *American Journal of Epidemiology* 2005; **162**(3): 267-78.

6. *The practice effect of cognitive assessment should be considered as a covariate in the statistical analysis.*

OUR RESPONSE: We strongly agree that practice effect is an important consideration in longitudinal studies of cognitive performance. We reran analyses including practice effect (using a dichotomous indicator showing whether it was a participant’s first round of cognitive testing as has been done previously in studies using ELISA^{3,4}) and found that results were unchanged. We have revised the manuscript to include this analysis in the Supplemental materials, and have also added the following text to the Methods and Results:

Methods, page 22, paragraph 2: “In further analyses, we examined the impact of adjusting for practice effect, body mass index and self-reported hearing difficulty on the results [...]”

Results, page 10, paragraph 1: “Adjustment for practice effect (Tables S9-S10), body mass index (Tables S11-S12), or hearing difficulty (Tables S13-S14) did not change the results.”

References

3. Bloomberg M, Brocklebank L, Hamer M, Steptoe A. Joint associations of physical activity and sleep duration with cognitive ageing: longitudinal analysis of an English cohort study. *The Lancet Healthy Longevity* 2023; **4**(7): e345-e53.

4. Bloomberg M, Dugravot A, Dumurgier J, et al. Sex differences and the role of education in cognitive ageing: analysis of two UK-based prospective cohort studies. *The Lancet Public Health* 2021; **6**(2): e106-e15.

7. *Did the authors consider the potential effects of some medications for chronic diseases on cognitive function?*

OUR RESPONSE: We agree that some medications could potentially confound associations between lifestyle and cognitive function; unfortunately, we did not have consistent information on medications to be able to include medication use as a confounder in our models. Therefore, we opted to acknowledge that medication could have an effect on cognition but did not include medication use in the analyses as this would have added further complexity to the analyses and interpretation of results. We have now included this as a limitation:

Discussion, page 11, paragraph 2: “Due to lack of data, we could not account for factors such as medication use which might confound associations between lifestyle and cognitive function. As such, residual confounding might limit causal inference [...]”

8. *Details regarding the number of participants lost or deceased during the 15-year follow-up and any differential effects across lifestyles are essential. Further insight into how these cases were handled in the analysis is needed.*

OUR RESPONSE: Thank you for pointing this out; we agree that this information is very important to include to understand differences in follow-up durations between lifestyles. While all respondents included in the analysis must have participated in the baseline wave (wave 2), after this point respondents have intermittent patterns of participation, where they may miss waves but still be interviewed at a later wave. In response to this comment, we have added a new table to the manuscript (Table 2) showing the sample size and median follow-up duration in each lifestyle and have also included a table in the Supplemental materials showing the number of participants still in the analytic sample at each wave in each lifestyle (Table S3). We have also revised the Results section to include the median follow-up duration by lifestyle in the main text:

Results, page 6, paragraph 3: “The number of participants with each lifestyle is reported in Table 2, with the number of participants remaining in the analytic sample at each study wave reported in Table S3. Median follow-up durations were similar for all lifestyles (between 7.5 and 10 years; Table 2), with the exception of two lifestyles: 1) smoking, no-to-moderate alcohol consumption, less-than-weekly MVPA, and less-than-weekly social contact (median=6 years, IQR=0-11) and 2) non-smoking, no-to-moderate alcohol consumption, less-than-weekly MVPA, and less-than-weekly social contact (median=6 years; IQR=0-11).”

Loss to follow-up is accounted for by using linear mixed models, which handle data missing-at-random⁵ by assuming that participants who are lost to follow-up have the same cognitive trajectory as those participants who have the same covariate values but with complete follow-up.

References

5. Berrington A, Smith P, Sturgis P. An overview of methods for the analysis of panel data. 2006.

9. *How was dementia diagnosed, and were the diagnostic criteria were consistent across both cohorts?*

OUR RESPONSE: Dementia was based on self-report of a clinical diagnosis. As such, criteria or procedures for diagnosis may differ slightly between countries. This was one reason that we also excluded individuals with cognitive scores suggesting cognitive impairment; though self-report is not necessarily a reliable means for dementia ascertainment, we were nonetheless able to exclude individuals with cognitive impairment regardless of whether they reported dementia diagnosis. We have now revised the Methods section to indicate that dementia ascertainment was based on self-report of clinical diagnosis:

Methods, page 18, paragraph 3: “To reduce the impact of prodromal dementia on behaviour, participants reporting having received a dementia diagnosis at any interview during the follow-up period [...] were excluded from analyses.”

10. *Participants who did not smoke or had no-to-moderate drinking habits been reported to have more chronic diseases. The study should address how this result was factored into the analysis.*

OUR RESPONSE: The finding that those who have smoking and drinking habits that are not compliant with recommendations are also less likely to have chronic conditions is most likely because individuals diagnosed with chronic conditions are more likely to quit smoking and to drink in moderation to manage their health. We have taken this into account by adjusting for chronic conditions in the models, and have revised the Methods section to make this clear:

Methods, page 19, paragraph 2: “Covariates were selected on the basis of previous evidence of associations with lifestyle and cognitive function and were ascertained by self-report. [...] Chronic conditions were ascertained based on self-report of clinical diagnosis and included high blood pressure, diabetes, cardiovascular conditions (including heart disease and stroke), cancer, lung disease, high cholesterol, and psychiatric conditions.”

Methods, page 20, paragraph 4: “We first fitted models adjusted for sociodemographic and socioeconomic covariates and chronic conditions to examine independent associations between each of the behaviours (smoking, alcohol consumption, MVPA, and social contact) and memory and fluency decline over 10 years. [...] We then fitted similar models to examine differences in memory and fluency decline over 10 years between the recommendation compliant lifestyle (non-smoking, no-to-moderate alcohol consumption, weekly MVPA, and weekly social contact) and the other lifestyles.”

11. *Given the last follow-up was in 2019/2020, it would be relevant to discuss the potential influence of the COVID-19 pandemic on the cognitive function of older adults in the present study.*

OUR RESPONSE: We agree that this is a very interesting research question, but as the follow-up period in this study concludes before the beginning of the pandemic (with the last data collection in March 2020), we are unfortunately unable to explore it further in this manuscript.

12. *The author concluded that associations between lifestyle and cognitive decline primarily depended on whether participants reported smoking, and examining behaviours independently or in indices does not fully capture associations between lifestyle and cognitive decline. It would be valuable for the authors to provide independent analysis results for the four health behaviors and discuss their relevance alongside the current findings.*

OUR RESPONSE: Thank you for this suggestion; we agree that examining independent associations between each behaviour and cognitive decline would add additional context to the results. We have now added these analyses to the manuscript, discussed the implications, and included a new table in the Supplemental materials (Table S6):

Methods, page 20, paragraph 4: “We first fitted models adjusted for sociodemographic and socioeconomic covariates and chronic conditions to examine independent associations between each of the behaviours (smoking, alcohol consumption, MVPA, and social contact) and memory and fluency decline over 10 years. These models included *time* (years since baseline), age at baseline in years (denoted age_0), the interaction between *time* and age_0 (denoted $time \times age_0$), all covariates, each behaviour, and the interaction between each behaviour and *time* (denoted behaviour $\times time$). Age_0 was centred at 65 years, the mean in the pooled analytic sample. After

determining that memory had a non-linear association with time, we also included $time^2$ and behaviour $\times time^2$ terms in the memory model only.”

Results, page 7, paragraph 2: “After adjustment for covariates, non-smoking and no-to-moderate alcohol consumption were independently associated with slower decline in both memory and fluency (Table S6). Non-smokers had memory scores that declined 0.08 SD (0.05 to 0.12) less than smokers over 10 years ($p < 0.001$) whilst those reporting no-to-moderate alcohol consumption declined 0.04 SD (0.01 to 0.06) less than those reporting heavy alcohol consumption ($p < 0.01$). Non-smokers’ fluency scores declined 0.08 SD (0.05 to 0.11) less than smokers ($p < 0.001$), and those reporting no-to-moderate alcohol declined 0.03 SD (0.01 to 0.06) less than those reporting heavy alcohol consumption ($p = 0.01$). There were no independent associations between memory decline and MVPA ($p = 0.58$) or social contact ($p = 0.46$), or between fluency decline and social contact ($p = 0.35$). There was weak evidence of an association between weekly MVPA and faster fluency decline (difference in decline over 10 years [less-than-weekly-weekly MPVA] = -0.03 SD [-0.05 to 0.00]; $p = 0.04$).”

Discussion, page 12, paragraph 3: “Physical activity is thought to be protective against cognitive decline and dementia in part due to its ability to function as a buffer against ageing-related neuronal loss, and strengthen compensatory mechanisms employed against ageing-related neurodegeneration;³² however, there is also evidence that associations between physical activity and dementia may be due to the effects of preclinical or prodromal dementia on physical activity rather than a causal effect of physical activity on dementia.³³ This is consistent with our finding that recommendation-compliant physical activity was not associated with slower rate of cognitive decline in this cognitively-healthy study population, although this result also could have been due to overreporting of physical activity. [...] Finally, social activity and engagement may promote neuroprotection by moderating associations between brain atrophy and cognitive function; social bonding might also moderate the association between perceived stress and cognitive function.³⁴ While we did not find an independent association between social contact and cognitive decline, when considered with other behaviours, social contact appeared to contribute to reductions in cognitive decline.”

Reviewer #2

1. *This paper is clear and well-written, and presents an original analysis of the association between different lifestyles and cognitive decline. I do have some reservations about the methods, the depth of the analyses, and conclusions drawn, that I think should lead to some revisions.*

OUR RESPONSE: Thank you for the positive feedback. We hope that our revisions will address your reservations.

2. *In the abstract and elsewhere, non-to moderate alcohol intake is referred to as healthier. But this is not borne out by the results shown in Table 1, and therefore ‘healthier’ is misleading.*

OUR RESPONSE: Thank you for pointing this out; we agree that referring to ‘healthier’ and ‘less healthy’ behaviours could be misleading, as we intended to refer to compliance with WHO recommendations for cognitive health, not to whether a behaviour is associated with fewer chronic conditions in our study population. We have removed ‘healthier’ and ‘less healthy’ in the abstract

and throughout the manuscript, and revised to refer to compliance with recommendations where appropriate, for example:

Abstract, page 2: “The reference lifestyle comprised not smoking, no-to-moderate alcohol consumption, weekly moderate-plus-vigorous physical activity, and weekly social contact.”

Methods, page 16, paragraph 2: “To facilitate interpretation of the results, behaviours were dichotomised into categories corresponding to ‘more recommendation compliant’ and ‘less recommendation compliant’ behaviour, based on WHO and United States guidelines.^{27,39}”

Methods page 17, paragraph 2: “The ‘recommendation compliant lifestyle’ refers to participants reporting non-smoking, no-to-moderate alcohol consumption, weekly MVPA, and weekly social contact.”

- 3. Smoking is the dominant factor, as shown by the authors. I think that once this has been shown, then more analyses are required to explore this further. In particular, the reference group of non-smokers will comprise past smokers (including recent ones) and never smokers. These are not likely to fit the same risk profile. I suggest that the smoking variable should be in 3 categories: never, past and current.*

OUR RESPONSE: Thank you for this suggestion. We agree that further analysis of the role of smoking is warranted given the results of the study. While sample size consideration precludes us from examining smoking in three categories in the lifestyle analyses, in response to this comment, we have now included examination of smoking in three categories as part of an analysis examining independent associations of each behaviour with cognitive decline to further elucidate the role of smoking (see the Supplemental materials, Table S6). We found that the difference in 10-year cognitive decline between never smokers and current smokers compared with former smokers and current smokers was very similar (for memory: 0.09 [95% CI 0.05 to 0.12] for former smokers vs. current smokers and 0.08 [0.04 to 0.12] for never smokers vs. current smokers; for fluency: 0.07 [0.03 to 0.11] for former smokers vs. current smokers and 0.09 [0.06 to 0.12] for never smokers vs. current smokers). As such, we feel categorising smoking into three groups in the main analysis is not necessary. We have included this analysis in the Methods and Results sections as follows:

Methods, page 22, paragraph 2: “In further analyses, we examined [...] independent associations between smoking and cognitive decline when smoking was split into three categories (never smokers, former smokers, and current smokers).”

Results, page 7, paragraph 2: “After adjustment for covariates, non-smoking and no-to-moderate alcohol consumption were independently associated with slower decline in both memory and fluency (Table S6). Non-smokers had memory scores that declined 0.08 SD (0.05 to 0.12) less than smokers over 10 years ($p < 0.001$) [...]. Non-smokers’ fluency scores declined 0.08 SD (0.05 to 0.11) less than smokers ($p < 0.001$) [...].”

Results, page 10, paragraph 1: “There were also negligible differences in associations between smoking and memory or fluency decline when current smokers were compared with former smokers or never smokers (Table S6).”

- 4. Addressing reverse causation. The authors have tried to minimise bias due to reverse causation but I do not think they have gone far enough. In particular, updating the exposure*

at every wave will introduce reverse causation bias, since some of those who change their behaviour might be caused to do so by incipient cognitive changes. I suggest that a further analysis is required which uses baseline exposures and does not update them. It is also not possible to say that reverse causation will not be affecting the results, as the follow-up time was about a decade. The best you can hope for, is that it is minimised.

OUR RESPONSE: We apologise for the confusion. We have already done as the reviewer suggests. The exposures are not updated at each wave; they are drawn from the baseline wave. We have clarified the Methods section to make this explicit:

Methods, page 15, paragraph 3: “The four behaviours considered were smoking, alcohol consumption, physical activity, and social contact, which were assessed at the baseline wave of the present study (wave 2 in both SHARE and ELSA) and were not updated during follow-up.”

We agree completely that it is not possible to eliminate bias due to reverse causation and have already indicated as such in the Discussion section, as well as the measures we have taken to reduce the likelihood that reverse causation is driving the results we observe:

Discussion, page 11, paragraph 1: “As prodromal dementia might influence behaviour years before diagnosis,²³ we reduced the likelihood of reverse causation driving the results we observed by focussing on cognitive decline rather than cross-sectional cognitive performance and by excluding participants with suspected cognitive impairment or who reported dementia diagnosis.”

- 5. Imputation of missings. I have some doubt over whether it also introduces bias to impute missings using the outcomes as well as the exposure. In any case, this could be avoided if you use baseline exposure.*

OUR RESPONSE: Please see our response to comment 4. We have already used the baseline exposure in our analyses as the reviewer suggests, and as such have avoided this source of bias.

- 6. Reporting non-significant differences between groups. I think the results should be revised, to make it clearer which differences are actually statistically significantly different. When looking at the confidence intervals that overlap in the tables, I would not think that many of the subgroups are significantly different from each other – except the smoking.*

OUR RESPONSE: Thank you for this suggestion; we acknowledge that the inability to distinguish which pairwise comparisons between lifestyles are statistically significant is a limitation of the way we have chosen to present the results. We chose to compare all lifestyles to a recommendation compliant reference lifestyle (please see Figures 1 and 2 in the manuscript) rather than examining pairwise comparisons because presenting 120 pairwise comparisons of lifestyles in text is not feasible. In response to this comment, we have revised the manuscript to remove instances where we have inappropriately implied statistically significant differences between subgroups. We have also revised the manuscript to include p-values in text, for example:

Results, page 8, paragraph 3: “In general, participants reporting smoking and weekly social contact also had faster memory decline than the reference lifestyle, but these differences did not always reach statistical significance. Participants reporting smoking, heavy alcohol consumption, less-than-weekly MVPA, and weekly social contact declined -0.09 SD (-0.19 to 0.00) more over 10 years than the reference lifestyle and those with weekly MVPA declined -0.08 (-0.18 to 0.02) more, but

these differences did not reach statistical significance ($p=0.06$ and $p=0.12$ respectively). Participants who reported smoking, no-to-moderate alcohol consumption, less-than-weekly MVPA, and weekly social contact declined -0.11 (-0.21 to -0.02) more over 10 years than the reference lifestyle ($p=0.02$). Participants who reported smoking, no-to-moderate alcohol consumption, weekly MVPA, and weekly social contact had no difference in memory decline compared with the reference lifestyle [difference in decline over 10 years= 0.00 [-0.10 to 0.10]; $p=0.96$].”

7. *Omission of BMI. The rationale given for the omission of BMI was not convincing. There is strong evidence that BMI is a risk factor for dementia (see studies by Kivimaki et al and by the Million Women Study team) after the removal of early years of follow-up. BMI has a stronger association with dementia than physical activity. In addition, it is not a strong argument to say that BMI is on the causal pathway from physical activity to cognitive decline, since BMI is much more strongly influenced by energy intake than by physical activity.*

OUR RESPONSE: We agree that BMI is likely to be more influenced by energy intake than by physical activity and in response to this comment have run a sensitivity analysis adjusting for BMI. The results after adjusting for BMI have not changed from the original results. We have included two new tables in the Supplemental materials (Tables S11-S12) and revised the Methods and Results as follows:

Methods, page 22, paragraph 2: “In further analyses, we examined the impact of adjusting for practice effect, body mass index and self-reported hearing difficulty on the results [...]”

Results, page 10, paragraph 1: “Adjustment for practice effect (Tables S9-S10), body mass index (Tables S11-S12), or hearing difficulty (Tables S13-S14) did not change the results.”

We have also removed the argument for omitting BMI as suggested:

Discussion, page 11, paragraph 2: “The inclusion of four behavioural factors may also limit causal inference; notably, we were unable to include sleep or diet though both behaviours may be relevant to cognitive ageing,^{25,26} as these data were not available in SHARE. Consideration of body mass index might partially account for dietary choices; the results were nonetheless unchanged after taking body mass index into account.”

8. *Discussion and interpretation of the results.*
line 348, line 425 and concluding paragraph – I disagree that there is a complex role for these factors. The results presented suggest that it is simple, in that only smoking really makes a big difference to cognitive performance. Therefore efforts should be made to encourage cessation and efforts do not need to be directed at the other factors. I do not see that there has been clear evidence presented here that the other factors investigated here could compensate.

OUR RESPONSE: We apologise for the lack of clarity. We have revised the manuscript to indicate we are referring to the *smoking, no-to-moderate-alcohol consumption, weekly MVPA, weekly social contact* lifestyle, which shows no difference in memory decline compared with the reference lifestyle. In the results for fluency decline, we also see that non-smoking lifestyles with at least one other healthy behaviour also do not differ significantly from the reference lifestyle. In these

instances, other behaviours compensated for the smoking behaviour. We have also revised the sentence identified by the reviewer to better summarise the preceding paragraph as suggested.

Discussion, page 10, paragraph 2: “The key finding arising from this examination of healthy lifestyle and 10-year cognitive decline is that associations between lifestyle and cognitive decline primarily depended on whether participants reported smoking. For those who reported current smoking, cognitive decline was generally faster than the reference lifestyle, except for those with recommendation compliant alcohol, MVPA, and social contact habits. By contrast, cognitive decline was generally similar between all non-smoking lifestyles, regardless of alcohol, physical activity, or social contact. Taken together, the results suggest an important role of smoking habits in shaping cognitive decline.”

9. *line 402 and line 430 – There is evidence that the reported association between physical activity and dementia is biased by reverse causation (see studies by the Whitehall II group and by the Million Women Study group).*

OUR RESPONSE: We agree that there is likely to be reverse causation at play in the association between physical activity and cognitive function. In our examination of independent associations between the behaviours and cognitive decline, we saw no evidence of an association between weekly MVPA and slower memory or fluency decline (Table S6). We have now included consideration of this reverse causal relationship in the Discussion sections as follows:

Discussion, page 12, paragraph 3: “Physical activity is thought to be protective against cognitive decline and dementia in part due to its ability to function as a buffer against ageing-related neuronal loss, and strengthen compensatory mechanisms employed against ageing-related neurodegeneration;³² however, there is also evidence that associations between physical activity and dementia may be due to the effects of preclinical or prodromal dementia on physical activity rather than a causal effect of physical activity on dementia.³³ This is consistent with our finding that recommendation-compliant physical activity was not associated with slower rate of cognitive decline in this cognitively-healthy study population, although this result also could have been due to overreporting of physical activity.”

Reviewers' Comments:

Reviewer #1:

Remarks to the Author:

The authors have addressed all my concerns, and I have no more questions.

Reviewer #2:

Remarks to the Author:

Thank you for taking into account my comments and adjusting your analyses where necessary.
The manuscript is much clearer and I have no further comments.